# Modification Tests to Optimize Compaction Quality Control of Granite Rockfill in Highway Embankments

**DOI:** 10.3390/ma13010233

**Published:** 2020-01-05

**Authors:** Evelio Teijón-López-Zuazo, Ángel Vega-Zamanillo, Miguel Ángel Calzada-Pérez, Luis Juli-Gándara

**Affiliations:** 1Construction and Agronomy Department, Zamora Polytechnical School, Viriato Campus, University of Salamanca, 49022 Zamora, Spain; 2Department of Transportation and Projects and Processes Technology, Civil Engineering Technical School of Santander, University of Cantabria, Los Castros Avenue, 39005 Santander, Spain; vegaa@unican.es (Á.V.-Z.); calzadam@unican.es (M.Á.C.-P.); luis.juli@alumnos.unican.es (L.J.-G.)

**Keywords:** granite, rockfill, quality control, compaction, wheel-tracking test, topographic settlement, plate bearing test, in-situ density

## Abstract

Particle size can be a problem in terms of rockfill compaction control methods, with little practical development of these techniques. The necessary fieldwork and laboratory tests were carried out to develop new quality control procedures for granite rockfill. This involved the revision of certain tests like the wheel-tracking or topographic settlement tests. More than 1100 in-situ density and moisture content measurements were performed for this research. In addition, more than 220 topographic settlements and 250 wheel-tracking carriage tests were carried out. The quality control processes were completed with 24 plate bearing tests. The results of granite rockfills were classified according to their use in the different areas of core or crown. Possible evidence of statistical correlations between compaction control tests was identified. An analysis of variance (ANOVA) was performed. When testing proved relationships between them, the replacement of one by the other was evaluated by deduction. Finally, the study suggests new procedures for compaction quality control of granite rockfill for its application in core and crown.

## 1. Introduction

In linear infrastructures, the quality control of rockfill is carried out using tests that cannot effectively evaluate the compaction process. Sopeña [1] indicates that there are no reference values for topographic settlements. Zhong et al. [2] developed a practical system for field engineers to conduct precise automatic online entire-process monitoring of compaction parameters that could overcome the disadvantages of conventional methods, which are easily influenced by human behavior and often lack adequate management. This system proved effective in the Nuozhadu dam and can, therefore, be applied to other civil works.

Teijón-López-Zuazo et al. [3] argue that pit gradings involving the weighing of different fractions of heavy rocks are not practical. Wheel-tracking tests are usually effective in normal compaction conditions, whereas plate bearing tests require the diameter of the element to be five times the maximum size of the aggregate. Likewise, nuclear density gauging methods are limited by large particle size and layer thicknesses above 30 cm. Although the modified Proctor test is the reference value for the degree of compaction, its main disadvantage is the substitution of fines when using a 20 mm sieve. In rockfills, less than 30% passing a 20 mm sieve means that there is a minimum of 70% substitute material, which leads to the conclusion that the test is not well represented. Substitution procedures such as in-situ soil density determination using the sand method are not representative either since such test is performed on soils where maximum sizes are below 50 mm.

Mazari and Nazarian [4] proved that quality (defined as achieving adequate layer modulus) is weakly associated with achieving density. Density and moisture measurements can be used as process control items, modulus-based measurements being used for quality acceptance. The influence depth of the light weight deflectometer is affected by layer thickness and by certain functional parameters such as the applied load and the plate diameter of the device. Therefore, moduli design can be estimated either using empirical relations or resorting to the classification of common local geomaterials.

For Fernández et al. [5], the lack of progress in compaction control justifies control by a procedure using established test sections. Even though there is a broad variety of rocks, this research is limited to the slate family, and its results were classified according to use in the different parts of the fill (core or crown). Subsequently, testing for possible correlations between the in-situ dry density test, wheel-tracking test, topographic settlement, and load-bearing test of soil was carried out. The statistical processing of the results yielded dependence relations. By using only representative tests, unnecessary pauses, and the purchasing of costly equipment were avoided. Instructions were also redefined in an attempt to establish efficient thresholds.

### 1.1. Igneous Rock (Granite)

According to Fernández et al. [5], granites meet the stability criteria to be successfully used in rockfill. The compressive strength of granitic rocks is highly variable, depending on the degree of alteration. Granites are classified into high resistance (80–100 MPa), medium resistance (40–50 MPa), and low resistance (20–25 MPa).

Brauns and Kast [6] indicate that rockfills are granular masses of solid fragments of sound rock. In the major portion of cases, the rock was disintegrated to a varying extent, resulting in a material compacted fill.

Akkurt et al. [7] obtain linear attenuation coefficients (µ) via mass attenuation coefficients (µ/φ), calculated using the XCOM code, which uses the chemical properties of a mixture of materials providing the database at photon energies from 1 keV to 100 GeV. The variation in the physical and mechanical properties of the rocks with the attenuation coefficients has been investigated, finding that such coefficients increase with increasing values of unconfined compressive strength (UCS). Thus, there is a positive correlation between the attenuation coefficient and UCS. The use of igneous rocks can provide important protection against radiation. There is a linear relationship between radiation shielding properties and certain physical and mechanical properties. Oteo [8] associates the requirements of materials not only with the concept of plasticity but also with granulometry and variation of optimum dry density.

The Technical Guide for Embankment Construction [9] classifies rocks according to the special behavior of their materials, which are, in turn, classified by their geological name. Such classification consists of six types of rock, known as the R_6_ group, among which are magmatic rocks like granites.

### 1.2. Compaction Control

The specifications required for the working procedure were established: Technical characteristics of the machinery to be used; the methods for the excavation, loading, transportation, and extension of stone material; layer thickness; compaction procedures; number of roller passes; procedures for the achievement of optimum moisture; tests implementing the same method with similar materials; and, finally, the possibility of increasing compaction assessment through post-compaction irrigation.

Oteo [8] indicates that a material made up of altered granite boluses should be used as rockfill. The digging, transport and arrangement require a specific study, and the control system must also be established, since the classic Proctor test hardly serves as a reference where such heterogeneous materials are concerned. The Spanish standard specifications [10] that the second/first moduli ratio yielded by the plate bearing test of soils (k) should be below 2.2 must be reconsidered since such limit was traditionally established for fine soils, which are very different from rockfill. For this reason, another criterion that allows for higher k values when Ev_2_ increases is proposed. In soils with large particle sizes, control tests using wheel-tracking and plate bearing tests are believed to be more appropriate. As regards compacted rockfill control, plastic density and geophysical methods are considered to be the best to obtain density. Radioactive isotope density can involve specific problems in the case of rocks that lack fine fractions since the dimensions of their particles do not allow the introduction of the gamma emitter into the ground. Although backscattering measurement is possible, the area of influence is smaller. The fact that, in this technique, the measurement is taken from the most superficial area, where compaction energy is higher, means that it yields higher values than thickness measurements and should, therefore, be correctly correlated with other control tests.

Niu et al. [11] described and characterized particle breakage that is important in predicting the behavior of coarse-grained soil as weathered granite. Especially, the effect of field compaction condition (such as the thickness of loose paving layer and number of vibratory rolling passes) significantly affected the particle breakage characteristics of weathered granite. The weathered granite showed obvious particle breakage characteristics under weak compaction effect and low stress levels and over-compaction could result in a decrease in the degree of compaction of a certain thickness of loose paying layer filled with weathered granite as subgrade fillings.

Wan-Huan et al. [12] found that the proposed method can estimate the SWCCs (soil water characteristic curve) of soils with different initial dry densities by considering initial porosity. In the wheel-tracking test, the seat is measured at ten points that are separated by a distance of 1 m, before and after the standard carriage passes. This test should be correlated with the plate bearing test, and the average settlement should not exceed 3 mm.

Kutzner [13] followed problems and questions that may arise during the course of geotechnical work on a dam project and explain investigations of construction materials. It includes an overview of the analysis used in advanced embankment dam engineering. This means in practice that the designer has to supervise the construction, with standards that request strict control of impounding.

Several experiments conducted by Fernández et al. [5], based on highway test sections, showed how plate bearing soil tests yield dispersed results so that the value of the k ratio increases at the same time as the modulus of the second loading stage (Ev_2_).

Zhong et al. [14] argued that the conventional quality control method of core rockfill dam construction exhibits difficulty controlling compaction parameters accurately or ensuring construction quality. They, therefore, established the timely monitoring indexes and control criteria of compaction processes by considering the characteristics and quality requirements of high core rockfill dam construction. A real-time monitoring system is provided to realize the precise automatic online entire-process monitoring of compaction parameters, including compaction pass, rolling trajectory, running speed of roller, vibration status and rolled pavement thickness.

Navarro et al. [15] reported encouraging results suggesting that granite fines (GF) are within an acceptable range to be used for the construction of embankment fills. Likewise, the analysis of the mechanical behavior of granite carried out by Liu et al. [16], showed a strong dependency on the confining pressure, so that increasing it significantly improved the load-bearing capacity and resistance of the rock specimen.

Zhang et al. [17] proposed a roller-integrated acoustic wave detection technique for rockfill materials. Based on Lamb’s problem and an infinite baffle piston radiation acoustic field model, a relationship model between the sound compaction value (SCV) and the dry density of the natural gravel materials (NGM) was established. This technique is an effective tool for compaction quality control of rockfill materials and has great potential for further applications.

Zhang et al. [18], analyzed the variation law of the displacement field, stress field, and plastic zone of embankment body reinforced by dynamic compaction with different energy levels and the optimal compaction energy by means of numerical simulations and field tests in view of the high filling height and a large amount of soil and rock in the high-filled embankment. The study showed that the displacement field and the stress field are redistributed after applying single-point compaction, and the volume of the shear plastic zone increases. The optimal number of slams for high-filled granular soil is seven times, and the effective depth of dynamic compaction is 4.5 m.

Sun et al. [19] tested 75 × 75 × 87 cm crushed rock samples under vertically cyclic loadings. Parameters were obtained using three coarsely crushed rock samples with initial grain sizes of 16–40, 25–50, and 50–80 mm, the results showing that cyclic vibration loadings can cause breakage and abrasion of particles and their edges in the coarsely crushed rock layer and that particles also tend to be rounded and non-angular. This results in a rearrangement of particles and a decrease in particle size, along with an increase in fine content in the coarsely crushed rock layer, thus reducing its porosity. Compared with the initial average porosity before cyclic loading, final average porosity reduction rates in the three crushed rock samples after cyclic loading at 18,000 cycles reach 6.53%, 7.45%, and 8.08% for initial grain sizes of 16–40, 25–50, and 50–80 mm, respectively. García et al. [20] compare different compaction control methods, also analyzing the granular sub-ballast that forms the sub-basin of railway lines. However, Ev_2_ does not provide information on the degree of compaction, leading to the conclusion that other criteria using Ev_1_ are more appropriate.

## 2. Materials and Methods

The materials were obtained from the A-66 Spanish highway, Cáceres (N)–Aldea del Cano section, which is also where the in-situ tests were performed.

### 2.1. Materials

All the tests were carried out during the quality control process, including initial identification and definition studies and subsequent control and adjustment during execution. They were applied to granite rockfills of more than 1,100,000 m^3^. Thus, Table 1 shows the sampling and average values of exploratory boreholes.

Rock cores correspond to different depths (2.60–11.20 m). Samples were classified into weathering grades II and III according to the degree of weathering of the granitic rock, the lowest alteration grade corresponding to the deepest sample (11.20 m). There is evidence of high compressive strength, with an average value of 144.9 kp/cm^2^. Dry densities yield values between 2.487–2.600 g/cm^3^, and moisture content is between 0.2% and 1.4%. Rock quality design (RQD) and rock mass rating (RMR) also score high, with average values of 91 and 72, according to rock hardness.

### 2.2. Methods

#### 2.2.1. Laboratory and Field Tests

This section describes the control methods and resulting specifications for compaction control according to the satisfactory results obtained using experimental sections. More than 1100 in-situ density and moisture measurements, UNE 103900 [21], 160 modified Proctor, 250 wheel-tracking tests, UNE 103407 [22], 220 topographic settlements and 24 plate bearing tests (Φ 762 mm), UNE 103808 [23], were carried out during this research.

Initially, the compaction control test procedures were revised in order to verify their effectiveness, modifying the procedures of the wheel-tracking and topographic settlement tests. The core includes foundation and backfills, and the crown consists of the top, transition layers and structure transitions.

The measurement points were obtained through the wheel-tracking test using a tape measure attached to two poles. The measurement element on which to place the topographic milestone consists of a set of welded metal frames commonly known as “H” because of its cross-linked arrangement. Measurements were taken by placing the metal device on each measurement point, before and after the loaded carriage passes. There were 10 measurement points spaced 1 m apart from each other and aligned to the left side of the truck. This means a reduced testing length when compared to the 100–200 m test length that is usual in compaction batches. Conversely, in order to make the test more representative, peg dispositions were modified, spacing the measurement points every 10 m. The rut to the left of the carriage proved equal to that of the right, the influence of the driver’s weight was insignificant due to the vehicle’s suspension performance. However, there were other deadweights such as the fuel tank that allows both sides to balance out, so that it was preferable to arrange the pegs into two rows, one per carriage roll. Finally, two points were checked using the filling profile. Thus, the final layout of the 10 measurement points was two rows with five measurement points each, separated every 10 m by two measurement points, as shown in Figure 1. This new arrangement allowed the assessment of 40 m, against the initial 10 m. Additionally, two measurements per section helped to compensate for any possible errors resulting from a single measurement of a heterogeneous material.

The leveling pegs could be quickly rearranged using edge stakes every 20 m. They should be inside the layer, at the same surface level. An initial leveling of the slabs was also required. Leveling using metal pegs was not necessary to position the measuring device “H” on each point since the topographic focus was placed directly on the metal pegs. This added speed to the test by not having to transport the “H” and, most importantly, it prevented inaccuracies resulting from measuring with the milestone resting on the ground, thus ensuring millimeter precision. The twin wheels were run over the alignment pegs by guiding the carriage from its front axle. Passing the front wheels through the middle of the slab ensures symmetry. The next step was to put the metal support into place, well centered on the measurement point and above the twin wheels. Homogeneous support of the “H” structure in the rut was guaranteed using captive screws that allow regulation of the length of the transversal frame, adapting it to the wheel impression. Once the leveling nail of the “H” was vertical on the point of the peg, readings with a topographic level on the “H” were conducted. The values of the depression caused by the passage of the truck were obtained by subtracting the height of the metal template constant. The value of the rut was calculated as the average mean of the 10 points (δ_m_). This value is called the degree of compaction index. A new limit of 4 mm was proposed, and extreme values deviating from the average mean were avoided, as established in the criteria for the revised test. Therefore, it was possible to reject up to three measurements while obtaining the mean. The topographic leveling was carried out with millimeter precision, although when the average of the depressions was obtained, a tenth of a millimeter was preserved to maintain the precision and differentiation of measurements. In this way, any possible extreme erroneous observations were reduced, alongside possible heterogeneities or instrumental errors that could affect the test, such as speed differences in carriage passes, the trajectory of the reference wheels, etc., that are difficult to control while the test is in process. The assistance of a specialized operator was necessary to guide the carriage driver so that the wheels could pass through the center of the pegs, facilitating the symmetry of the twin wheels with respect to the leveling point at a constant speed (similar to that of a man walking). Hence, the advantages of the revised procedure are as follows:The length that the revised test covers five times that of the initial test and offers two measurements for the same section.Reduction of leveling errors by means of a fixed point over the leveling peg. Millimetric accuracy is guaranteed, avoiding ground distortion.Higher performance by reducing test times. The first measurements were made over the pegs without the need to move the heavy metal support.The dynamic effects of carriage acceleration and braking become minimized. The revised test ensures constant speed when the carriage passes over the pegs.Two measurements per section provide more thorough testing of the section than measuring a single point. When measuring in two parallel and independent ruts, any exceptional deficiency in one of them becomes corrected. In addition, second-order effects such as driver or fuel tank weight are excluded from the test.

Another significant control test is the topographic settlement. There is a standard procedure test in which the last roll pass must be under 1% of the thickness of the compacted layer. This settlement must be measured after the first roll pass. Figure 2 shows the experimental verification of how this criterion has easily exceeded the settlement threshold at the first pass. Additionally, there are undefined indicators in the arrangement of the dots, their spacing, and how they were measured.

Likewise, it does not set a new origin for the repetition of seats in case of non-compliance, maintaining the original settlement after the first pass, which could give rise to ambiguities such as layer rejection before compaction finishes. Hence, it is appropriate to revise this control method. As shown in the previous figure, topographic control must reach the stabilization of values by increasing the number of passes when compaction is completed. In addition, the measurement points must be defined, proposing the same arrangement as in the wheel-tracking test. Therefore, extreme values deviating from the mean in such test were excluded.

In-situ density measurements were performed using nuclear gauges, whose high performance and high speed of operation and delivery of results enabled testing for possible correlations with the rest of the compaction control tests. The plate bearing test of soil requires a plate diameter to be at least five times the maximum material size, so a 762 mm diameter plate was chosen. The general specifications applied to the quality control of the granite rockfills are summarized in Table 2.

#### 2.2.2. Statistical Estimation

Multilinear adjustment models with dependency relationships have been researched. These dependency relationships allowed for the dependent variables to be evaluated without the need to run them. All models include well-defined validity zones.

Since there was a large number of lots, the Kolmogorov–Sminornov test was used instead of the Shapiro–Wilk test. Sometimes the Shapiro–Wilk test is used as a contrast. Specifically, 125 compaction lots at core and 40 at crown were processed. Every compaction lot was subjected to a minimum of two tests. The independent variables used for each control lot were generated and introduced into the IBM SPSS statistical program. More than 12 variables were analyzed, and independent variables with low absolute values in the Student’s t-test were discarded because they were not significant for statistical adjustment. Accordingly, the independent variables used were the following:d: Average lot density (g/cm^3^).s: Average settlement between last and first roller pass (mm).h: Average wheel impression after test carriage (mm).Ev_1_: First vertical modulus of the plate bearing test (MPa).Ev_2_: Second vertical modulus of the plate bearing test (MPa).k: Relationship between second and first moduli of the plate bearing test (Ev_2_/Ev_1_).

In general, a coefficient of determination value of R^2^ ≥ 0.70 was the cutoff point for a strong relationship. An ANOVA analysis of variance yielded the sums of squares, the degrees of freedom, and Levene’s F statistic.

## 3. Results

Possible linear correlations between two compaction control tests were researched extensively. Test results were collected in compaction lots so that the batches with the two trials analyzed were represented as points. Because of the linear nature of the adjustment studied, it is not relevant to define variables as dependent or independent.

### 3.1. Core Granite Rockfill

Possible correlations between 125 compaction lots were evaluated. No relationship between the following tests was found: Wheel-tracking–topographic settlement test, density–first modulus of the plate bearing test (Φ 762 mm), wheel-tracking–first modulus of the plate bearing test (Φ 762 mm), wheel-tracking–second modulus of the plate bearing test (Φ 762 mm), topographic settlement test–second modulus of the plate bearing test (Φ 762 mm), wheel-tracking–relationship between second and first moduli of the plate bearing test (Ev_2_/Ev_1_) and topographic settlement–relationship between second and first moduli of the plate bearing test (Ev_2_/Ev_1_).

#### 3.1.1. Relationship between In-Situ Density and Topographic Settlement

As shown in Figure 3, the correlation between in-situ density and topographic settlement is high. The distribution is inversely proportional, with lower in-situ density values corresponding to higher settlement values.

As shown in Table 3, Pearson correlation coefficient is high, ρ = 0.844, standard error is low, Se = 0.3250 MPa, and there is a high coefficient of determination, R^2^ = 0.713. In other words, there is a high correlation associated with low dispersion.

Table 4 shows the ANOVA parameters. Levene’s test is clearly significant, Sig. = 0.001 with F = 24.813, which means that the homoscedasticity criterion is not met since the variances are significantly different. The variables show a strong dependency relationship.

Student’s t-test values are significant and, as shown in Table 5, topographic settlement contributes significantly in the second modulus of the plate bearing test (Φ 762 mm).

The expression of the adjustment line is:s = 20.399 − 7.792 d R^2^ = 0.713(1)
where s is a topographic settlement in millimeters and d is density in grams per cubic centimeter. Function domain values are between (1.96 ≤ d ≤ 2.18) and (3.1 ≤ s ≤ 5.4).

#### 3.1.2. Relationship between Wheel-Tracking Test and First Modulus of the Plate Bearing Test

As shown in Figure 4, there is a high correlation between the wheel-tracking test and the first modulus of the plate bearing test (Φ 762 mm), with inverse proportionality. In this case, high wheel-tracking test values correspond with low first modulus plate bearing test values.

Table 6 shows a high Pearson coefficient value, ρ = 0.866, associated with low dispersion. The coefficient of determination, R^2^ = 0.749, validates a variance of 75%. The standard error is only 12.7337 MPa.

The ANOVA parameters are shown in Table 7. Levene’s test was significant Sig. = 0.001 with a value of F = 23.894. Therefore, the null hypothesis of homoscedasticity is rejected, since variances are significantly different.

The t-test in Table 8 shows high values of 11.880 and −4.888, both significant (Sig. < 0.05).

The wheel-tracking test predicts the first vertical modulus of the plate bearing test. Besides, according to linear regression coefficients, the fit between the wheel-tracking test and the first modulus of the plate bearing test (Φ 762 mm) is:Ev_1_ = 127.155 − 26.062 h R^2^ = 0.749(2)
where Ev_1_ is the first modulus of the plate bearing test in megapascals and h is the wheel-tracking test in millimeters. Function domain is between (50 ≤ Ev_1_ ≤ 120) and (0.5 ≤ h ≤ 3.5).

### 3.2. Crown Rockfill

Forty compaction lots were tested for possible correlation, finding no relationship between the following tests: In-situ density—topographic settlement, in-situ density—first modulus of the plate bearing test (Φ 762 mm), topographic settlement—first modulus of the plate bearing test (Φ 762 mm), topographic settlement—second modulus of the plate bearing test (Φ 762 mm) and topographic settlement—relationship between second and first moduli of the plate bearing test.

#### 3.2.1. Relationship between Topographic Settlement Test and Wheel-Tracking Test

As shown in Figure 5, there is a high correlation between topographic settlement and wheel-tracking test. Thus, the association is directly proportional, higher values of the wheel-tracking test corresponding to higher topographic settlements.

Table 9 shows a high Pearson correlation coefficient value, ρ = 0.832, low standard error Se = 0.6977 MPa, and a high coefficient of determination R^2^ = 0.693. In other words, there is a high correlation associated with low dispersion.

ANOVA offers the parameters shown in Table 10. Levene’s test is clearly significant, sig = 0.010 with F = 13.518. Therefore, the homoscedasticity criterion is not met, since the variances are significantly different. The variables clearly have a strong dependency relationship.

Student’s t-test values are significant. As shown in Table 11, there is a significant contribution of the wheel-tracking test in the topographic settlement.

The expression of the adjustment line is:s = 2.945 + 0.965 h R^2^ = 0.693(3)
s being the topographic settlement test and h being the wheel-tracking test, both reflected in millimeters. The function domain uses the intervals of (0.5 ≤ s ≤ 3.0) and (0.5 ≤ h ≤ 3.0).

#### 3.2.2. Relationship between Wheel-Tracking Test and First Modulus of the Plate Bearing Test

As shown in Figure 6, there is a possible inversely proportional correlation between the wheel-tracking test and the first modulus of the plate bearing test (Φ 762 mm). In this case, high values for the wheel-tracking test correspond with low values of the first modulus of the plate bearing test, although a larger sample would be necessary to completely ascertain it.

Table 12 shows a high Pearson coefficient value of ρ = 0.996, which is associated with low dispersion. The coefficient of determination R^2^ = 0.993 validates a variance of 99.3%. The standard error is only 3.9812 MPa.

ANOVA yields the parameters in Table 13. Levene’s test was significant sig = 0.055 with a value of F = 134.189. Therefore, the null hypothesis of homoscedasticity is rejected, since variances are significantly different.

T-test yields high values, 24.096 and −11.954, both significant (sig = 0.026 and 0.055). Likewise, the wheel-tracking test predicts the first vertical modulus of the plate bearing test, although it is necessary to complete the ascertainment with a bigger sampling.

#### 3.2.3. Relationship between Wheel-Tracking Test and Second Modulus of the Plate Bearing Test

As shown in Figure 7, there is a high correlation between the wheel-tracking test and the second modulus of the plate bearing test (Φ 762 mm), with inverse proportionality. In this case, there are high values for the wheel-tracking test corresponding to low values for the second modulus plate bearing test, although a larger sample would be necessary to completely ascertain it.

Table 14 shows a high Pearson coefficient value of ρ = 0.971, which is associated with low dispersion. The coefficient of determination R^2^ = 0.942 validates a variance of 94.2%. The standard error is only 22.9296 MPa.

ANOVA offers the parameters in Table 15, with a value of F = 16.333. Therefore, the null hypothesis of homoscedasticity is rejected, since variances are significantly different.

T-test yields high values, 9.949 and −4.041, both significant. Moreover, the wheel-tracking test predicts the second vertical modulus of the plate bearing test, although further analysis using a larger sample is required to completely ascertain it.

#### 3.2.4. Relationship between Wheel-Tracking Test and Relation between Second and First Modulus of the Plate Bearing Test (Ev_2_/Ev_1_)

As shown in Figure 8, there is a high correlation between the wheel-tracking test and the relationship between the second and first moduli of the plate bearing test (Ev_2_/Ev_1_). Thus, the association is directly proportional, higher values of the wheel-tracking test corresponding with a higher relationship between the second and first moduli of the plate bearing test, although a larger sample is required to fully assert this.

Table 16 shows a high Pearson coefficient value of ρ = 0.996, which is associated with low dispersion. The coefficient of determination R^2^ = 0.992 validates a variance of 99.2%. The standard error is only 0.0392 MPa.

ANOVA offers the parameters in Table 17. Levene’s test was significant sig = 0.000 with a value of F = 120.333. Therefore, the null hypothesis of homoscedasticity is rejected, since the variances are significantly different.

Thus, the wheel-tracking test predicts the relationship between the second and first modulus of the plate bearing test (Ev_2_/Ev_1_), although further analysis using a larger sample is required to completely ascertain it.

## 4. Discussion

The granite rockfill in core results was grouped into a significance matrix, as can be observed in Table 18, which shows the combinations of analyzed tests alongside their coefficients of determination. When not representative, the numerical value was replaced by ns (nonsignificant). Certain obvious relationships were not considered.

In-situ density correlates with the topographic settlement. First modulus of the plate bearing test (Φ 762 mm), wheel-tracking, and second modulus of the plate bearing test (762 mm) proved to have a strong relationship. A revised control method was designed for topographic settlement and plate bearing test (Φ 762 mm).

Additionally, the granite rockfill in crown results was grouped into a significance matrix, as shown in Table 19, which illustrates the combinations of analyzed tests with their coefficients of determination. When not representative, the numerical value was replaced by ns (not significant). Some relationships were not considered because of their being obvious. In some cases, however, further analysis using a larger sample would be necessary to completely ascertain the results.

The wheel-tracking test density correlates with the topographic settlement. The moduli of the plate bearing test (Φ 762 mm) and the wheel-tracking test (762 mm) proved to have a strong relationship. A revised control method was designed for in-situ density, topographic settlement, and plate bearing tests (Φ 762 mm).

The wheel-tracking test lacks precision in that the distance tested is very short, and the measurements are taken at soil surface level. Thus, the wheel-tracking test has been revised in an attempt to remedy these shortcomings. The improved wheel-tracking test includes defining the use of pegs, more measurement points, and the assessment of longer distances. Even so, the strong correlation between this improved version and the plate bearing test means that, according to the findings, the plate bearing test (Φ 762 mm) could replace the improved wheel-tracking test.

The traditional topographic settlement test usually measures the first and last compaction roller passes. In the improved topographic settlement test, measurements were also taken twice, but the first time the compaction roller passes were not measured. Instead, the two measurements were taken after the compaction roller passes the second to last time and after the last compaction roller pass.

Even after the improvement of the topographic settlement test, the correlation between it and the plate bearing test (Φ 762 mm) proved so strong that the plate bearing test could easily replace it as well. Although both the topographic settlement test and the wheel-tracking test were significantly improved, the plate bearing test correlates so strongly with them that the plate bearing test (Φ 762 mm) could replace both of them.

Test analyses provide promising results supporting the possibility of using sizes larger than fine grain soils in rockfill at the crown level.

## 5. Conclusions

In some cases, the maximum size of the particles in rockfill restricts the accurate execution of several tests, including in-situ density definition, modified Proctor, plate bearing test, topographic settlement, and wheel-tracking test. For this reason, the wheel-tracking test and the topographic settlement test were revised and improved to optimize their results. The statistical processing of the control tests has simplified the quality control procedure in granite rockfills. Some major conclusions drawn from the findings include:A new wheel-tracking test procedure was proposed, with measurement points in two rows of five measurement points each, with a separation of 10 m between each other.The topographic settlement control method was reviewed, considering its limitations. Measurements were taken after the second to last compaction roller pass and after the last compaction roller pass.The wheel-tracking test correlates strongly with other compaction control tests and can, therefore, be replaced to avoid redundant results.For core granite rockfill, adjustments have been made to replace the wheel-tracking test with the topographic settlement test and the plate bearing test (Φ 762 mm).For crown granite rockfill, there is a high correlation between the wheel-tracking test and the topographic settlement and the plate bearing test.In conclusion, the improved tests proposed for the quality control are the in-situ density test, topographic settlement, and the plate bearing test (Φ 762 mm).

## Figures and Tables

**Figure 1 materials-13-00233-f001:**
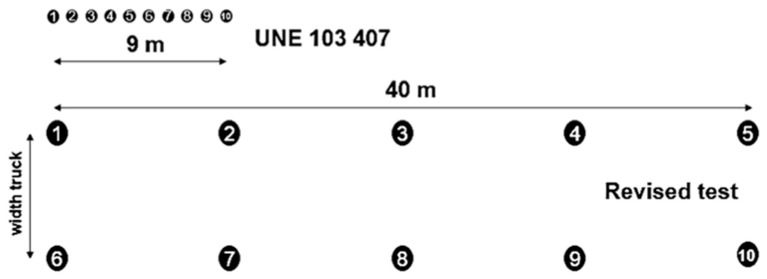
Revised wheel-tracking test measurement points.

**Figure 2 materials-13-00233-f002:**
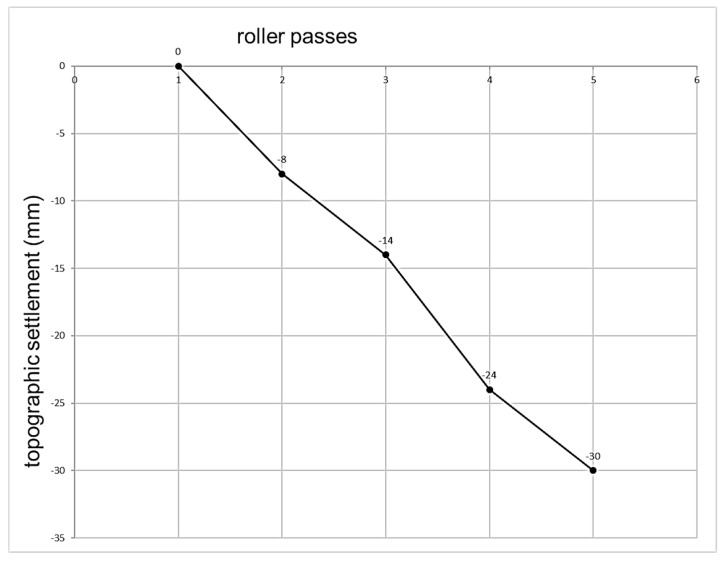
Topographic settlement. Experimental section granite rockfill 1000 mm.

**Figure 3 materials-13-00233-f003:**
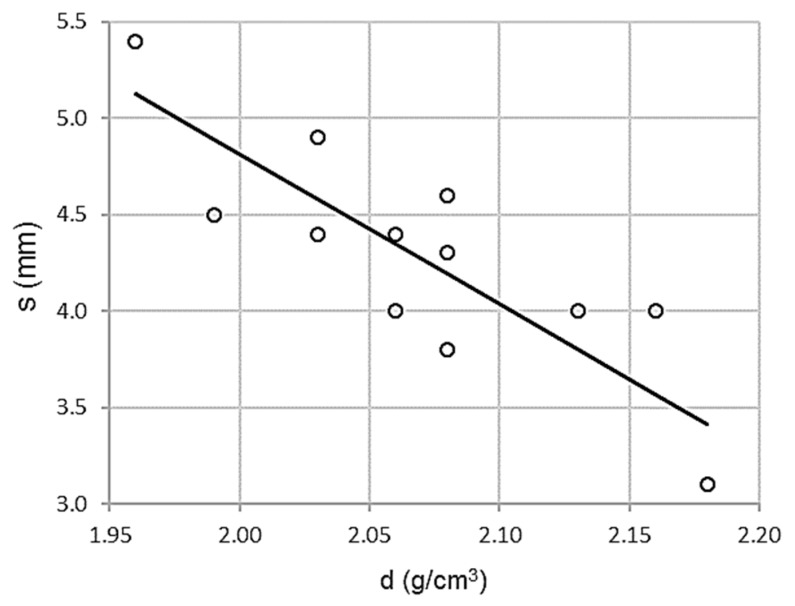
Scatterplot for in-situ density and topographic settlement test.

**Figure 4 materials-13-00233-f004:**
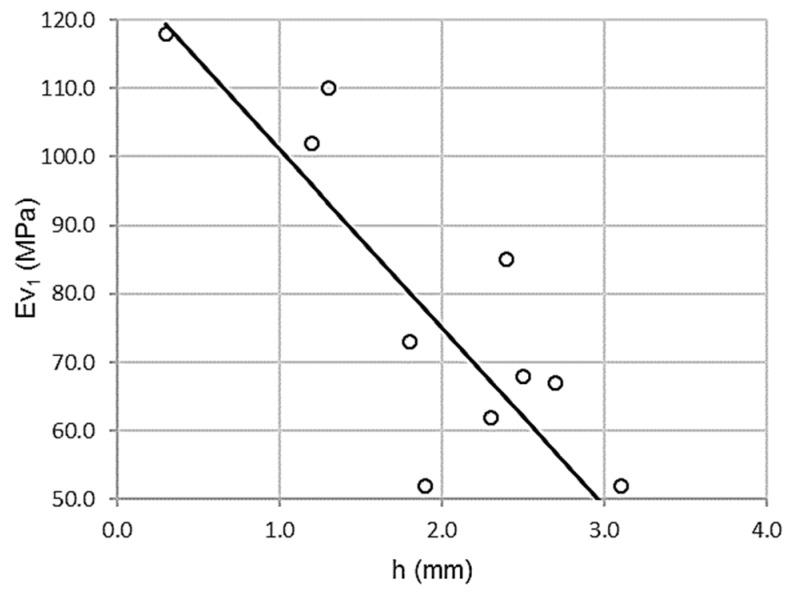
Scatterplot for wheel-tracking test and first modulus of the plate bearing test (Φ 762 mm).

**Figure 5 materials-13-00233-f005:**
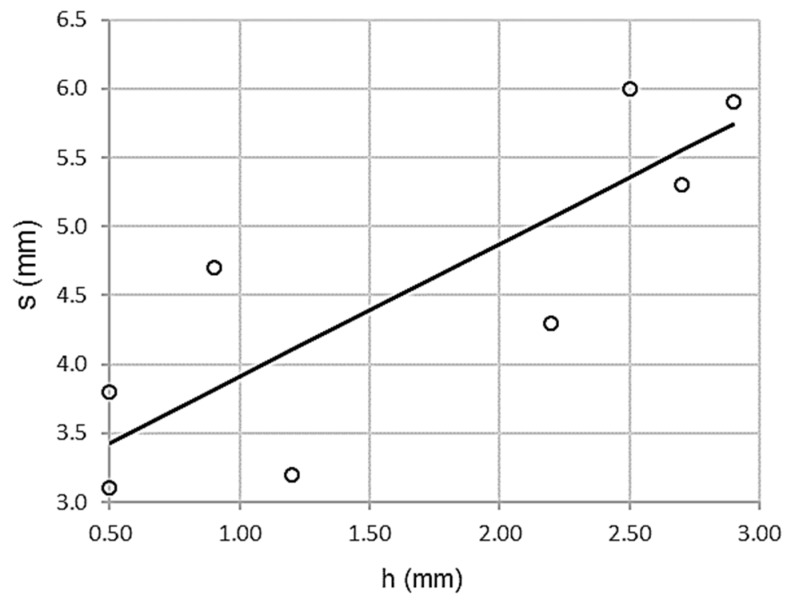
Scatterplot for wheel-tracking and topographic settlement tests.

**Figure 6 materials-13-00233-f006:**
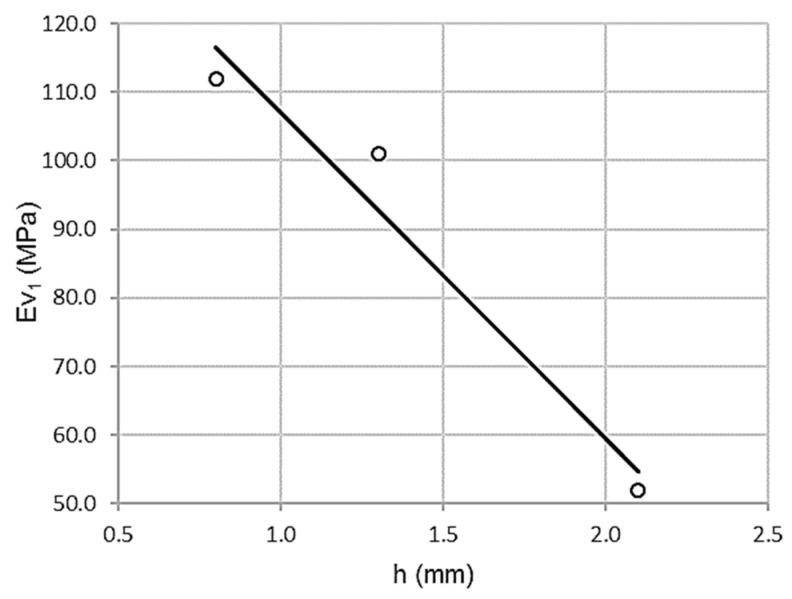
Scatterplot for wheel-tracking test and first modulus of the plate bearing test (Φ 762 mm).

**Figure 7 materials-13-00233-f007:**
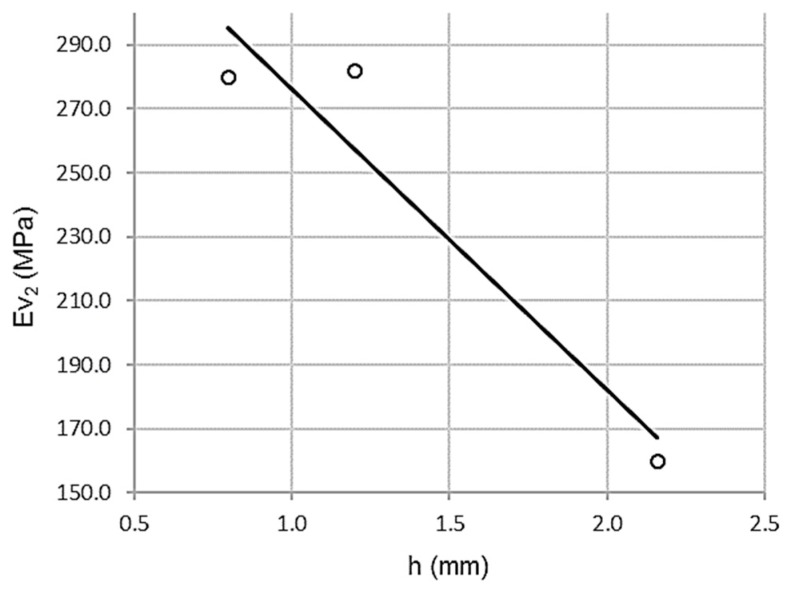
Scatterplot for wheel-tracking test and second modulus of the plate bearing test (Φ 762 mm).

**Figure 8 materials-13-00233-f008:**
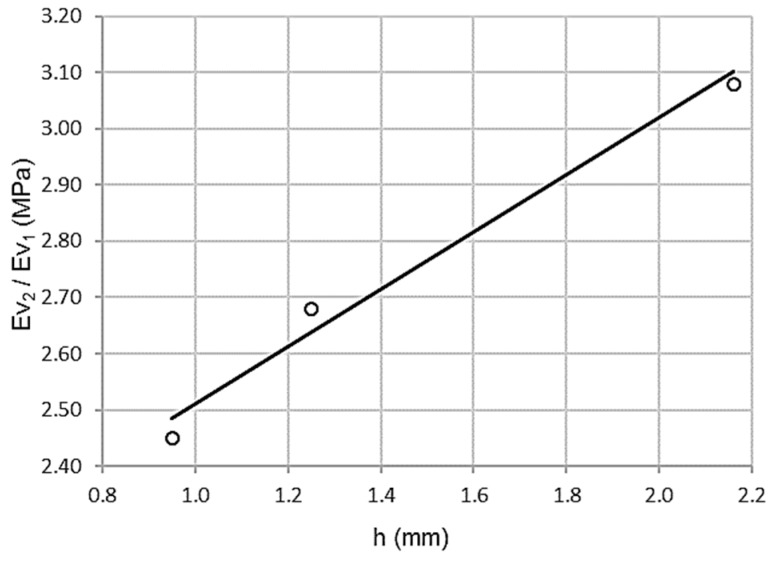
Scatterplot for wheel-tracking test and relation between second and first moduli of the plate bearing test (Ev_2_/Ev_1_).

**Table 1 materials-13-00233-t001:** Granitic rock characteristics.

Ref.	Depth (m)	Weathering Grade	Bulk Density (g/cm^3^)	Dry Density (g/cm^3^)	Moisture Content (%)	Compressive Strength (kp/cm^2^)	RQD	RMR
SD-1-1	2.60–2.90	III	2.522	2.487	1.4	119.5	100	100
SD-1-2	3.30–5.00	III				150.0	85	62
SD-1-3	7.15–7.45	III	2.605	2.600	0.2	160.2	100	65
SD-1-4	10.20–11.20	II				150.0	80	62
		III	2.564	2.544	0.8	144.9	91	72

**Table 2 materials-13-00233-t002:** Specifications suggested for rockfills.

		Settlement (mm)		Modulus (MPa)		k (Ev_2_/Ev_1_)
Zone	Compaction Degree (%)	Wheel	Topographic	Ev_1_	Ev_2_	
core		≤4.0	≤5.0	≥50.0		<3.2
crown	98.0	≤3.0	≤5.0		≥120.0	<3.6
	not required					

**Table 3 materials-13-00233-t003:** Determination coefficients: In-situ density and topographic settlement.

Summary Model			
R	R^2^	R^2^ Adjusted	Standard Error
0.844 ^a^	0.713	0.684	0.3250

^a^ Predictors: Constant, d (g/cm^3^).

**Table 4 materials-13-00233-t004:** Analysis of variance: In-situ density and topographic settlement.

ANOVA ^a^					
Model	Sum of Squares	Degrees of Freedom	Quadratic Average	F	Sig.
regression	2.621	1	2.621	24.813	0.001 ^b^
sampling error	1.056	10	0.106		
total	3.667	11			

^a^ dependent variable: s (mm). ^b^ predictors: (constant), d (g/cm^3^).

**Table 5 materials-13-00233-t005:** Linear regression coefficients: In-situ density and topographic settlement.

Coefficients ^a^					
Model	Nonstandard Coefficients		Standard Coefficients	T	Sig.
	B	Standard Error	Beta		
(constant)	20.399	3.237		6.303	0.000
d (g/cm^3^)	−7.792	1.564	−0.844	−4.981	0.001

^a^ dependent variable: s (mm).

**Table 6 materials-13-00233-t006:** Determination coefficients: Wheel-tracking test and first modulus of the plate bearing test.

Summary Model			
R	R^2^	R^2^ Adjusted	Standard Error
0.866 ^a^	0.749	0.718	12.7337

^a^ Predictors: Constant, h (mm).

**Table 7 materials-13-00233-t007:** Analysis of variance: Wheel-tracking test and first modulus of the plate bearing test.

ANOVA ^a^					
Model	Sum of Squares	Degrees of Freedom	Quadratic Average	F	Sig.
regression	3874.334	1	3874.334	23.894	0.001 ^b^
sampling error	1297.182	8	162.148		
total	5171.516	9			

^a^ dependent variable: Ev_1_ (MPa). ^b^ predictors: (constant), h (mm).

**Table 8 materials-13-00233-t008:** Linear regression coefficients: Wheel-tracking test and first modulus of the plate bearing test.

Coefficients ^a^					
Model	Nonstandard Coefficients		Standard Coefficients	T	Sig.
	B	Standard Error	Beta		
(constant)	127.155	10.703		11.880	0.000
d (g/cm^3^)	−26.062	5.332		−4.888	0.001

^a^ dependent variable: s (mm).

**Table 9 materials-13-00233-t009:** Determination coefficients: Wheel-tracking test and topographic settlement.

Summary Model			
R	R^2^	R^2^ Adjusted	Standard Error
0.832 ^a^	0.693	0.641	0.6977

^a^ Predictors: Constant, h (mm).

**Table 10 materials-13-00233-t010:** Analysis of variance: Wheel-tracking test and topographic settlement.

ANOVA ^a^					
Model	Sum of Squares	Degrees of Freedom	Quadratic Average	F	Sig.
regression	6.580	1	6.580	13.518	0.010 ^b^
sampling error	2.920	6	0.487		
total	9.500	7			

^a^ dependent variable: s (mm). ^b^ predictors: (constant), h (mm).

**Table 11 materials-13-00233-t011:** Linear regression coefficients: Wheel-tracking test and topographic settlement.

	Coefficients ^a^				
Model		Nonstandard Coefficients	Standard Coefficients	T	Sig.
	B	Standard Error	Beta		
(constant)	2.945	0.501		5.873	0.001
h (mm)	0.965	0.263	0.832	3.677	0.010

^a^ dependent variable: s (mm).

**Table 12 materials-13-00233-t012:** Determination coefficients: Wheel-tracking test and first modulus of the plate bearing test.

Summary Model			
R	R^2^	R^2^ Adjusted	Standard Error
0.996 ^a^	0.993	0.985	3.9812

^a^ Predictors: Constant, h (mm).

**Table 13 materials-13-00233-t013:** Analysis of variance: Wheel-tracking test and first modulus of the plate bearing test.

ANOVA ^a^					
Model	Sum of Squares	Degrees of Freedom	Quadratic Average	F	Sig.
regression	2126.837	1	2126.837	134.189	0.055 ^b^
sampling error	15.850	1	15.850		
total	2142.687	2			

^a^ dependent variable: Ev_1_ (MPa). ^b^ predictors: (constant), h (mm).

**Table 14 materials-13-00233-t014:** Determination coefficients: Wheel-tracking test and second modulus of the plate bearing test.

Summary Model			
	R^2^	R^2^ Adjusted	Standard Error
0.971 ^a^	0.942	0.885	22.9296

^a^ Predictors: Constant, h (mm).

**Table 15 materials-13-00233-t015:** Analysis of variance: Wheel-tracking test and second modulus of the plate bearing test.

ANOVA ^a^					
Model	Sum of Squares	Degrees of Freedom	Quadratic Average	F	Sig.
regression	8584.806	1	8584.806	16.333	0.154 ^b^
sampling error	525.600	1	525.600		
total	9110.407	2			

^a^ dependent variable: Ev_2_ (MPa). ^b^ predictors: (constant), h (mm).

**Table 16 materials-13-00233-t016:** Determination coefficients: Wheel-tracking test and relation between second and first moduli of the plate bearing test (Ev_2_/Ev_1_).

Summary Model			
R	R^2^	R^2^ Adjusted	Standard Error
0.996 ^a^	0.992	0.984	0.0392

^a^ Predictors: Constant, h (mm).

**Table 17 materials-13-00233-t017:** Analysis of variance: Wheel-tracking test and relationship between second and first moduli of the plate bearing test (Ev_2_/Ev_1_).

ANOVA ^a^					
Model	Sum of Squares	Degrees of Freedom	Quadratic Average	F	Sig.
regression	0.185	1	0.185	120.333	0.058 ^b^
sampling error	0.002	1	0.002		
total	0.187	2			

^a^ dependent variable: k. ^b^ predictors: (constant), h (mm).

**Table 18 materials-13-00233-t018:** Granite rockfill in core significance matrix.

Determination Coefficients (R^2^)						
Title	d (g/cm^3^)	h (mm)	s (mm)	Ev_1_ (MPa)	Ev_2_ (MPa)	k (Ev_2_/Ev_1_)
d (g/cm^3^)						
h (mm)		-				
s (mm)	0.713	ns	-			
Ev_1_ (MPa)	ns	0.749	ns	-		
Ev_2_ (MPa)	(*)	ns	ns		-	
k (Ev_2_/Ev_1_)	(*)	(*)	ns			-

ns: Non-significant (*) obvious relationships.

**Table 19 materials-13-00233-t019:** Granite rockfill in crown significance matrix.

Determination Coefficients (R^2^)						
Title	d (g/cm^3^)	h (mm)	s (mm)	Ev_1_ (MPa)	Ev_2_ (MPa)	k (Ev_2_/Ev_1_)
d (g/cm ^3^)	-					
h (mm)		-				
s (mm)	ns	0.693	-			
Ev_1_ (MPa)	ns	(**)	ns	-		
Ev_2_ (MPa)	(*)	(**)	ns		-	
k (Ev_2_/Ev_1_)	(*)	(**)	ns			-

ns: Non-significant (*) obvious relationships. (**) insufficient sample size.

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
