# Peer review of "Modification Tests to Optimize Compaction Quality Control of Granite Rockfill in Highway Embankments"

_materials, 2020, doi:10.3390/ma13010233_

Round 1
Reviewer 1 Report
This paper reports an evaluation of the quality control for the compaction of granite rockfill embankments. Although the topic is of interests and this research collected many valuable data, the paper is not well organized and difficult to follow. The research objectives are not clearly indicated in the introduction. The experiments involved are not clearly explained with sufficient details. If there are standard available for the experiments, it should be clearly indicated. If not, they should be provided with sufficient details so the readers can understand how they are carried out. The results can be significantly improved. The some of the conclusions are still what has been done in this study, but not supported directly by the results. The paper can be significantly improve after a proof reading by a professional editor. There are many grammar issues and ambiguous expression which are detrimental for the understanding.
Specific comments hope can help to improve the quality of this paper:
Line 11: “effectively” should be “effective” Line 14: meets over? Line 22: recommends change to “typically done” Line 37: what the igneous rock should be discussed separately? Just because this study used granite rock? Are the results of this tudy only applicable for granite rock? Line 47-48: what are the physical and mechanical properties? How can these properties be connected with the compaction and density? Line 57: what is “thickness layers”? do you mean layer thickness? Line 59: what do you mean by “increase of compaction evaluation” how the compaction evaluation can be increased? Do you mean improvement of the compaction evaluation? Line 64-66: recommend to show the equation and explain the parameters for better understanding. Line 67: what are the specific tests for the control? Line 101-102: the parameters of Ev1 and Ev2 are not defined. Recommend show details of equation and the definition of the parameters. Line 104: “the materials ….were done located at…” is confusing. Line 107: what are the “initial identification and definition studies?” Table 1. Typically the bulk density is a dry density just including the volume of the voids. If the dry density is not bulk, it means it's the density of solids, then the dry density should be larger than the bulk density. I guess the bulk density refers to wet density? They need to be defined. Line 116: the rock quality design (RQD) and the rock mass rating (RMR) need to be defined. Line 123-125: the UNE 10390 etc. are references, please follow consistent way for reference. Line 123-125: it looks there are many data collected, but in the figures there are much less data dot plotted. Does this mean there many data are not used in the correlation or data cannot match for the correlation, i.e. in-situ density are topographic settlements are not measured at the same location? If so, the quantity of data actually used in the analysis should be reported here, instead the data collected but actually not used in the analysis. Results can be reported with a more concise way. Some of the statistical results may not be included. Most of the time a report of the significance would be sufficient, unless the authors want to discuss other parameters. For example, line 324 “the t-test offers high values, 24.096 and -11.954, both signifiers.” What are the meaning of the 24.96 and -11.954? they are high values compare with what? Line 344-345: correlation between the wheel impression test and what? Figure 8: the captain indicate it correlation between second and first modulus (Evs/Ev1) of the plate load test, but the figure shows K and h. it needs to be explained.Line 398-400: this is what has been done in this study, but not conclusions can be supported directly but the results of this study. Same as line 403-404.
Author Response
Thank you very much for your attention and for your useful remarks. We are pleased to send a revised version of the article, included the improvements you kindly suggested. The technical part has been improved and the manuscript has been proffead, corrected and edited by a professional translator. We sincerely apreciate the opportunity to consider our work.
Thank you again for you attention, best regards

Reviewer 2 Report
Dear authors,
your article and mainly the findings of your research are interesting, however, there are some issues that, in my opinion, should be specified in more detail or changed. Also, the presentation and overall form (layout) of your paper is often very confusing and disturbing. Please, use the template and precisely follow the recommendation of the publisher.
Page 1: Article name: The name is hard to understand and its modification should be considered, e.g. Modification tests to optimize quality control in compaction of granite rockfill embankments for highways.
Abstract: It is known that the abstract should help readers remember key findings on a presented topic. The structure of the abstract can be as follows: 1) the research focus; 2) the research methods; 3) the results/findings of the research; 4) the main conclusions and recommendations. However, I did not find any abstract. Why?
Line 6: What is “… lineal infrastructures …”?
The chapter Introduction is very poor. It does not make any sense to an uninitiated reader. The methods that are mentioned via references should be introduced briefly and then the results of other authors should be mentioned. Try to re-write this part of the article, you can merge the first part with part 1.1 and 1.2. The aim of the presented article should be mentioned at the end of this chapter as well.
Line 29 and Line 38: Is “Fernández et al.” and “Fernández” the same reference, why do you use different referencing style?
Line 63: It is stated that “The Spanish standard specifications … “, since your article is focused on description of optimal quality control test method, I would recommend to consider requirements of more international standards or to compare the requirements of different standards as are e.g. ASTM, DIN, BSI etc.
Line 107 – 108: “The tests were carried out during quality control, as well as initial identification and definition studies, as well as subsequent control and adjustment during execution.” 1) During what type of quality control were the tests carried out, was it the initial test? 2) initial definition as well as definition studies – Is it a construction project phase? This sentence does not make any sense!
Line 123 – 125: “More than 1100 “in situ” density and moisture measurements, UNE 103900 [15], 160 modified Proctor, 250 wheel impression tests, UNE 103407 [16], 220 topographic settlements and 24 load plate tests (ɸ 762mm), UNE 103808 [17].”. Please, rewrite this sentence it is terrible. It should be as follows: More than 1100 “in situ” density and moisture measurements were carried out according to recommendations given by UNE 103900, etc.
I would recommend to rename the chapter Results to Results and Analysis and merge it with chapter Discussion. Otherwise, the chapter Discussion is quite poor and insufficient for this type of article.
Reference: The number of relevant references is in my opinion very insufficient. The described research area is nowadays very popular and therefore I assume that there are more relevant studies than those mentioned in references, e.g.: Laboratory Testing and Quality Control of Rockfill — German Practice by J. Brauns, K. Kast, or Earth and Rockfill Dams: Principles for Design and Construction by Christian Kutzner etc.
Author Response
Thank you very much for your attention and for your useful remarks. We are pleased to send a revised version of the article, including the improvements you kindly suggested. the technical part has been improved and the manuscript has been proofread, corrected and edited by a professional translator.
We sincerely appreciate the opportunity to consider our work. Thanking you again for your attention, best regards

Round 2
Reviewer 1 Report
Comments of reviewers were not sufficiently addressed (no responses to comments?). It is generally understood that the reviewers are not always correct (actually the authors understand the research better). But the authors’ points need to be justified to address the concern raised. A discussion will be helpful to address some comments especially when some comments were not reflected in the revised paper.
The English is significantly improved, but still some errors, such as line 16 “above than”, line 103 “The materials and in-situ tests were based on the A-66 Spanish highway” (The materials were evaluated and in-situ tests were based on….??”, line 108 “above 1100000 m3” is confusing, etc.
Reviewer 2 Report
Dear authors,
there are still too many errors that need to be considered:
1) Authors are still missing. Why?
2) The abstract is still missing. Why?
3) You did not consider this change. Why? - Line 29 and Line 38: Is “Fernández et al.” and “Fernández” the same reference, why do you use the different referencing style?
4) You did not consider this change. Why?
Reference: The number of relevant references is in my opinion very insufficient. The described research area is nowadays very popular and therefore I assume that there are more relevant studies than those mentioned in references, e.g.: Laboratory Testing and Quality Control of Rockfill — German Practice by J. Brauns, K. Kast, or Earth and Rockfill Dams: Principles for Design and Construction by Christian Kutzner etc.
References - the number of references is very poor for this type of article and journal. Please, look for more relevant sources!!!
5) The part Discussion is still very insufficient and unprofessional, as I said in my previous review. As I have already stated, you are discussing the results in the chapter results, so I'm suggesting to merge these to parts and rename the chapters. OR in chapter results, you could show only the data and then in chapter Discussion, point out the main findings.
Regards.
Round 3
Reviewer 1 Report
The comments have been sufficiently addressed.
Reviewer 2 Report
Dear, the article still has some flaws that need to be corrected. It is the first page of the article. Please edit this page according to the requirements of the journal.